# Correlations of intracranial pathology and cause of head injury with retinal hemorrhage in infants and toddlers: A multicenter, retrospective study by the J-HITs (Japanese Head injury of Infants and Toddlers study) group

Mihoko Kato[1], Masahiro Nonaka[2]*, Nobuyuki Akutsu[3], Ayumi Narisawa[4], Atsuko Harada[5], Young-Soo Park[6]

1 Department of Neurosurgery, Aichi Children's Health and Medical Center, Aichi, Japan, 2 Department of Neurosurgery, Kansai Medical University, Osaka, Japan, 3 Department of Neurosurgery, Hyogo Prefectural Kobe Children's Hospital, Hyogo, Japan, 4 Department of Neurosurgery, Sendai City Hospital, Miyagi, Japan, 5 Department of Pediatric Neurosurgery, Takatsuki General Hospital, Osaka, Japan, 6 Department of Neurosurgery, Nara Medical University, Nara, Japan

* nonakamasa65@gmail.com

## Abstract

### Introduction

In infants who have suffered head trauma there are two possible explanations for retinal hemorrhage (RH): direct vitreous shaking and occurrence in association with intracranial lesions. Which possibility is more plausible was examined.

### Material and methods

This multicenter, retrospective study reviewed the clinical records of children younger than four years with head trauma who had been diagnosed with any findings on head computed tomography (CT) and/or magnetic resonance imaging (MRI). Of 452 cases, 239 underwent an ophthalmological examination and were included in this study. The relationships of RH with intracranial findings and the cause of injury were examined.

### Result

Odds ratios for RH were significant for subdural hematoma (OR 23.41, p = 0.0004), brain edema (OR 5.46, p = 0.0095), nonaccidental (OR 11.26, p<0.0001), and self-inflicted falls (OR 6.22, p = 0.0041)

### Conclusion

Although nonaccidental, brain edema and self-inflicted falls were associated with RH, subdural hematoma was most strongly associated with RH.

**Data Availability Statement:** All relevant data are within the manuscript and its Supporting Information files.

**Funding:** The authors received no specific funding for this work.

**Competing interests:** Masahiro Nonaka and Young-Soo Park have written statements and appeared in court in child abuse cases both on the request of the prosecutor and the defense. Atsuko Harada has written statements and appeared in court in child abuse cases on the request of the prosecutor. This does not alter our adherence to PLOS ONE policies on sharing data and materials.

## Introduction

Retinal hemorrhage in infants with head trauma, especially in severe cases, may be caused by abusive behavior such as shaking [1–3]. Retinal hemorrhage in infants is thought to occur because, when infants are shaken, the eye's vitreous body is also shaken, resulting in a traction force on the retina [1]. This theory is the basis for the hypothesis that retinal hemorrhage in infants is caused by violent head shaking.

On the other hand, another theory suggests that the intracranial environment, such as intracranial hemorrhage or increased intracranial pressure, causes retinal hemorrhage. These include retinal vein congestion and leakage from increased intracranial pressure through the optic nerve sheath [4] or venous stasis associated with increased intracranial pressure [5].

To determine whether retinal hemorrhage is caused by an intracranial lesion or by a direct external force on the eye, 239 cases of infants and toddlers with records of ophthalmological examinations after head trauma were examined.

## Methods

This multicenter, retrospective study reviewed the clinical records of children under four years of age with head trauma who visited our facility from January 2014 through August 2020. The definition of head trauma in our study included cases in which the parent or guardian stated during the interview that the child had suffered a head injury, as well as cases in which there were findings on imaging that were thought to be due to head trauma. Furthermore, Patients with imaging findings, such as skull fractures or intracranial injuries of trauma who lacked caretaker histories of a traumatic event were included.

Patients with no imaging findings, patients with only extracranial findings, and patients with no apparent findings on imaging were excluded. Two university hospitals, two children's hospitals, and two general hospitals participated in this study, which made the data more similar to the actual situation in Japan [6, 7]. A total of 452 children were included in the study. The following information was extracted from the medical records: sex and age of the child, mechanism of injury, physical and neurological findings, radiological findings, presence of retinal hemorrhage, surgical procedures, notification to the child welfare center, temporary custody by the child welfare center, and criminal cases. Imaging studies reviewed in this study included computed tomography (CT) and magnetic resonance imaging (MRI). Imaging findings were reviewed and recorded by pediatric neurosurgeons at each institution. Information regarding the mechanism of injury was determined primarily from the medical history provided by the parents to the physicians and nurses. If there were any changes in the parents' explanation of the cause of the injury, they were noted verbatim, and the most recent description of the injury, as stated by the parents, was included in the study. Cases of suspected abuse were carefully reviewed by a child protection team composed of medical social workers, nurses, pediatricians, emergency physicians, and radiologists at each facility. If the team determined that abuse was likely, the cases were notified to the Child Guidance Center. In addition, cases in which abuse was strongly suspected were also reported to the police.

Of the 452 patients enrolled in J-HITS [6], those who did not undergo ophthalmological examinations were excluded, and 239 patients with fundal findings were included in this study. From the medical records, sex, age at injury, mechanism of injury, physical and neurological findings, radiological findings, presence of retinal hemorrhage, notification to the Child Guidance Center, temporary custody by the child guidance center, and confirmed criminal cases were extracted. Cases in which the caregivers confessed to abuse, or cases in which the caregivers did not admit to but were suspected of abuse and the child was placed in temporary custody by the Child Guidance Center, were classified as nonaccidental, and other cases

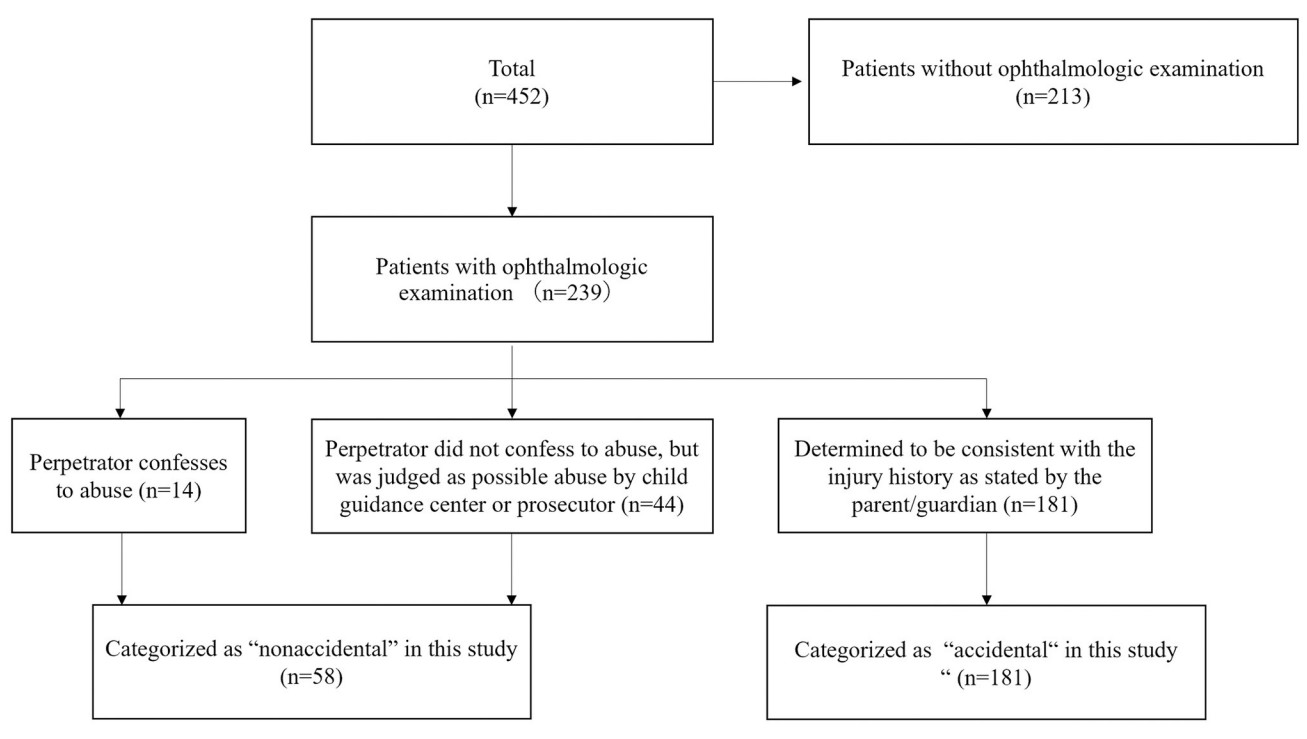

**Fig 1. Flow chart of patient selection.** Of the 452 registered cases, 294 without ophthalmological examinations were excluded. Of the remaining 239 cases, 58 were categorized as "nonaccidental" and 181 as "accidental".

were classified as accidental (Fig 1). The Child Guidance Center in Japan receives notifications of suspected cases of abuse at medical institutions, etc., and determines whether or not abuse actually occurred. The Child Guidance Center determines whether or not the child should be placed in temporary custody based on the child's history of visits to medical institutions and health checkups, whether or not the sibling has been abused, and the results of a diagnosis by a third-party doctor. The process and reasons for how the determination of whether or not abuse was made will not be communicated to the medical institution that notified of the abuse. In cases where abuse is strongly suspected, the child is almost always given temporary protection. On the other hand, cases that are determined to be unlikely to be abusive are not protected. Therefore, in this study, the cases that were temporarily protected by Child Guidance Center were judged as abuse, and those that were not protected were judged as accidents.

For the Accidental cases, the parent's description of the injury was listed, and the descriptions were further categorized into birth injury, motor vehicle accident, bicycle accident, fall from >2 m, falling while being held by a parent/caregiver, parents/caregivers dropped, self-inflicted fall, falling from a bed or sofa, other falls from <2 m, based on the content of the description. Abnormal findings on CT and MRI were confirmed and recorded by pediatric neurosurgeons at each institution. Self-inflicted falls occurs mainly when an infant falls backward and bruises the back of the head, mainly when trying to stand while holding onto other objects.

### IRB/Ethics committee approval and a statement regarding patient consent

The protocol for this study was approved by the Ethics Committee of Kansai Medical University (No. 2019232). Need for written patient consent was waived by the ethics committee

because data were deidentified. Institutional review board approval was obtained from all participants' institutions prior to submitting cases for this study.

## Statistical analysis

Statistical analysis was performed using JMP 14.2.0. Univariate and multivariate analyses were performed to examine the relationships between retinal hemorrhage and other factors. A univariate logistic model was used to compare each factor. If the p-value on univariate analysis was significant, the variable was included in the multivariate logistic regression model. Odds ratios (ORs) and 95% confidence intervals (CIs) were calculated.

## Results

Of the 452 patients enrolled in J-HITS, 239 underwent ophthalmological evaluation; retinal hemorrhage was observed in 85 patients (Table 1). A summary of the associations between retinal hemorrhage and findings and symptoms of head injury, including skull fracture, epidural hematoma, subdural hematoma, traumatic subarachnoid hemorrhage, cerebral contusion, cerebral edema, and seizure, is shown in Table 1.

Of the 85 cases with retinal hemorrhage, a subdural hematoma was found in 83 patients. Of the cases in which the parents confessed to abuse, two had retinal hemorrhages but no subdural hematoma. One of the cases had cerebral edema and subarachnoid hemorrhage, and the other had an epidural hematoma. Table 2 shows the association between the mechanism of injury and retinal hemorrhage.

Table 3 shows the injury mechanisms described by the parents in our study and the percentage of nonaccidental judgments for each. The presence or absence of retinal hemorrhages

**Table 1. Age, sex, and imaging findings of patients by the presence of retinal hemorrhage.**

|  |  | Retinal hemorrhage | | | Univariate analysis | |
|  |  | yes | no | total | odds ratio | p value |
|---|---|---|---|---|---|---|
| Age | under 6 mo. | 38 | 69 | 107 | 0.99 | 0.99 |
|  | over 7 mo. | 47 | 85 | 132 |  |  |
| Sex | male | 63 | 93 | 156 | 1.9 | 0.03 |
|  | female | 22 | 61 | 83 |  |  |
| Subdural hematoma | yes | 83 | 44 | 127 | **104** | **< .0001** |
|  | no | 2 | 110 | 112 |  |  |
| Brain edema | yes | 24 | 4 | 28 | **14.8** | **< .0001** |
|  | no | 61 | 150 | 211 |  |  |
| Epidural hematoma | yes | 1 | 43 | 44 | **0.03** | **0** |
|  | no | 84 | 111 | 195 |  |  |
| Subarachnoid hemorrhage | yes | 10 | 17 | 27 | 1.07 | 0.87 |
|  | no | 75 | 137 | 212 |  |  |
| Skull fracture | yes | 16 | 116 | 132 | **0.08** | **< .0001** |
|  | no | 69 | 38 | 107 |  |  |
| Brain Contusion | yes | 5 | 14 | 19 | 0.63 | 0.38 |
|  | no | 80 | 140 | 220 |  |  |
| Seizure | yes | 50 | 20 | 70 | 9.5 | 9.5 |
|  | no | 35 | 133 | 168 |  |  |
| Total |  | 85 | 154 | 239 |  |  |

Significant p-values are shown in bold.

**Table 2. Cause of injury of patients by the presence of retinal hemorrhage.**

| Cause of injury | | | Retinal Hemorrhage | | | Univariate analysis | |
|---|---|---|---|---|---|---|---|
| | | | yes | no | total | odds ratio | p value |
| Nonaccidental | | yes | 45 | 13 | 58 | **12.2** | **< .0001** |
| | | no | 40 | 141 | 181 | | |
| Birth injury | | yes | 1 | 2 | 3 | 0.9 | 0.94 |
| | | no | 84 | 152 | 236 | | |
| Traffic accidents | Motor vehicle accident | yes | 2 | 4 | 6 | 0.9 | 0.91 |
| | | no | 83 | 150 | 233 | | |
| | Bicycle accident | yes | 0 | 5 | 5 | N/A | N/A |
| | | no | 85 | 149 | 234 | | |
| Falls from >2m | | yes | 2 | 4 | 6 | 0.9 | 0.91 |
| | | no | 83 | 150 | 233 | | |
| Falls from <2m | Falling while being held by a parent/caregiver | yes | 2 | 17 | 19 | **0.19** | **0.03** |
| | | no | 83 | 137 | 220 | | |
| | Parents/caregivers dropped | yes | 0 | 44 | 44 | N/A | N/A |
| | | no | 85 | 110 | 195 | | |
| | Self-inflicted fall | yes | 16 | 10 | 26 | **3.34** | **0.01** |
| | | no | 69 | 144 | 213 | | |
| | Falling from a bed or sofa | yes | 7 | 17 | 24 | 0.72 | 0.49 |
| | | no | 78 | 137 | 215 | | |
| | Other falls from <2m | yes | 5 | 30 | 35 | **0.26** | **0.01** |
| | | no | 80 | 124 | 204 | | |

Significant p-values are shown in bold.

is also noted. The cases in which the parents did not fully explain the cause of the injury tended to be judged as abuse, while the cases in which the cause of the injury was fully explained tended to be judged as accidental. Retinal hemorrhage was present in 65.6% of self-inflicted falls.

Univariate and multivariate analyses were performed to analyze the factors related to retinal hemorrhage. Univariate analysis showed that the odds ratios of retinal hemorrhage were significantly higher for male sex (OR 1.9, p = 0,03), subdural hematoma (OR 103.75, p<0.0001), brain edema (OR 14.75, p<0.0001), seizure (OR 9.75, p<0.0001), nonaccidental (OR 12.2, p<0.0001), and self-inflicted falls (OR 3.34, p = 0.005). On the other hand, the odds ratios were significantly lower for epidural hematoma (OR 0.03, p = 0.0007), skull fracture (OR 0.076, p<0.0001), falling while being held by a parent (OR 0.19, p = 0.031), and other falls from <2 m (OR 0.26, p = 0.0072) (Table 4). Multivariate analysis that included factors found to be significant on the univariate analysis was then performed. The factors with significantly increased odds ratios were subdural hematoma (OR 23.41, p = 0.0004), brain edema (OR 5.46, p = 0.0095), nonaccidental (OR 11.26, p<0.0001), and self-inflicted falls (OR 6.22, p = 0.0041) (Table 4).

Self-inflicted falls, which were found to be significantly different in the multivariate analysis, have not been reported to be a factor associated with retinal hemorrhage. Therefore, we further examined the data in detail. Table 5 shows a detailed examination of the characteristics of self-inflicted falls by each facility. The results showed that self-inflicted falls were present in all facilities, ranging from 2.0–27.8% of the total number.

**Table 3. Parents' description of the injury mechanism and determination of nonaccidental or accidental injury in this study.**

| Cause of injury explained by caregivers | Total | All Determined accidental | All Determined nonaccidental | % of nonaccidental | Total | % of RH | With retinal hemorrhage Determined accidental | With retinal hemorrhage Determined nonaccidental | % of nonaccidental |
|---|---|---|---|---|---|---|---|---|---|
| Birth injury | 3 | 3 | 0 | 0 | 1 | 33.3 | 1 | 0 | 0 |
| Motor vehicle accident | 6 | 6 | 0 | 0 | 2 | 33.3 | 2 | 0 | 0 |
| Bicycle accident | 5 | 5 | 0 | 0 | 0 | 0 | 0 | 0 | 0 |
| Falls from >2 m | 7 | 6 | 1 | 14.3 | 2 | 28.6 | 2 | 0 | 0 |
| Falling while being held by a parent/caregiver | 25 | 19 | 6 | 24 | 6 | 24 | 2 | 4 | 66.7 |
| Parents/caregivers dropped | 45 | 44 | 1 | 2.22 | 1 | 2.22 | 0 | 1 | 100 |
| Self-inflicted fall | 32 | 26 | 6 | 18.8 | 21 | 65.6 | 16 | 5 | 23.8 |
| Falling from a bed or sofa | 25 | 24 | 1 | 4 | 8 | 32 | 7 | 1 | 12.5 |
| Other falls from <2 m | 35 | 35 | 0 | 0 | 5 | 14.3 | 5 | 0 | 0 |
| Unexplained convulsions | 20 | 2 | 18 | 90 | 15 | 75 | 1 | 14 | 93.3 |
| Unexplained coma | 7 | 1 | 6 | 85.7 | 7 | 100 | 1 | 6 | 85.7 |
| Other unexplained events | 6 | 4 | 2 | 33.3 | 2 | 33.3 | 0 | 2 | 100 |
| Confessed abuse | 14 | 0 | 14 | 100 | 10 | 71.4 | 0 | 10 | 100 |
| Other injury | 9 | 6 | 3 | 33.3 | 5 | 55.6 | 3 | 2 | 40 |
| Total | 239 | 181 | 58 | 24.3 | 85 | 35.6 | 40 | 45 | 52.9 |

**Table 4. Results of univariate and multivariable analyses to examine the relationships between retinal hemorrhage and various factors.**

| | | Univariate analysis Odds ratio | Univariate analysis p value (Prob>ChiSq) | Univariate analysis Lower 95% CI | Univariate analysis Upper 95% CI | Multivariable analysis Odds ratio | Multivariable analysis p value (Prob>ChiSq) | Multivariable analysis Lower 95% CI | Multivariable analysis Upper 95% CI |
|---|---|---|---|---|---|---|---|---|---|
| Age <6 months | | 0.99 | 0.99 | 0.58 | 1.7 | | | | |
| Male | | 1.9 | **0.03** | 1.05 | 3.36 | 1.36 | 0.53 | 0.53 | 3.51 |
| Subdural hematoma | | 103.75 | **< .0001** | 24.45 | 440.28 | 23.41 | **0.0004** | 4.04 | 135.69 |
| Brain edema | | 14.75 | **< .0001** | 4.91 | 44.3 | 5.46 | **0.0095** | 1.52 | 19.68 |
| Epidural hematoma | | 0.03 | **0.0007** | 0.0041 | 0.23 | 0.19 | 0.2 | 0.014 | 2.51 |
| Subarachnoid hemorrhage | | 1.07 | 0.87 | 0.47 | 2.46 | | | | |
| Skull fracture | | 0.076 | **< .0001** | 0.039 | 0.15 | 0.45 | 0.17 | 0.15 | 1.42 |
| Brain Contusion | | 0.63 | 0.38 | 0.22 | 1.8 | | | | |
| Seizure | | 9.5 | **< .0001** | 5.02 | 17.98 | 1.36 | 0.49 | 0.56 | 3.26 |
| Nonaccidental | | 12.2 | **< .0001** | 5.99 | 24.82 | 11.26 | **< .0001** | 3.72 | 34.03 |
| Birth injury | | 0.9 | 0.94 | 0.08 | 10.13 | | | | |
| Traffic accidents | Motor vehicle accident | 0.9 | 0.91 | 0.16 | 5.04 | | | | |
| | Bicycle accident | N/A | N/A | N/A | N/A | | | | |
| Falls from > 2m | | 0.9 | 0.91 | 0.16 | 5.04 | | | | |
| Falls from <2 m | Falling while being held by a parent | 0.19 | **0.031** | 0.044 | 0.86 | 2.27 | 0.48 | 0.23 | 22.45 |
| | Parents dropped | N/A | N/A | N/A | N/A | | | | |
| | Self-inflicted fall | 3.34 | **0.005** | 1.44 | 7.74 | 6.22 | **0.0041** | 1.78 | 21.68 |
| | Falling from a bed or sofa | 0.72 | 0.49 | 0.29 | 1.82 | | | | |
| | Other falls from <2 m | **0.26** | **0.0072** | 0.096 | 0.69 | 1.51 | 0.57 | 0.36 | 6.36 |

Significant p-values are shown in bold.

**Table 5. Number and characteristics of self-inflicted falls by each facility.**

| | | Facilities | | | | | | |
|---|---|---|---|---|---|---|---|---|
| | | **A** | **B** | **C** | **D** | **E** | **F** | **Total** |
| Total number of patients enrolled from each facility | | 58 | 13 | 33 | 51 | 18 | 66 | 239 |
| Self-inflicted falls (Parents/Caregiver's initial explanation) | | 12 | 2 | 6 | 1 | 5 | 6 | 32 |
| | % of Self-inflicted falls in total | 17.2 | 15.4 | 6.1 | 2 | 27.8 | 9.1 | 10.9 |
| | Nonaccidental | 2 (16.7%) | 0 (0%) | 4 (66.7%) | 0 (0%) | 0 (0%) | 0 (0%) | 6 (18.8%) |
| | Mean age (months) | 9.1 | 9 | 10.3 | 11 | 9.8 | 9.8 | 9.6 (range 5–15) |
| | Falling backward while standing holding onto something (Parent/Caregiver's initial explanation) | 9 (75.0%) | 2 (100%) | 4 (66.7%) | 1 (100%) | 3 (60.0%) | 5 (83.3%) | 24 (75.0%) |
| | Falling backward while sitting (Parent/Caregiver's initial explanation) | 1 (8.3%) | 0 (0%) | 2 (33.3%) | 0 (0%) | 2 (40.0%) | 1 (16.6%) | 6 (18.8%) |
| | Male | 10 (83.3%) | 2 (100%) | 6 (100%) | 1 (100%) | 3 (60.0%) | 5 (83.3%) | 27 (84.3%) |
| | Retinal hemorrhage | 6 (50.0%) | 2 (100%) | 4 (66.7%) | 1 (100%) | 3 (60.0%) | 5 (83.3%) | 21 (65.6%) |
| | Subdural hematoma | 7 (58.3%) | 2 (100%) | 5 (83.3%) | 1 (100%) | 4 (80.0%) | 6 (100%) | 25 (78.1%) |
| | Brain edema | 1 (8.3%) | 0 (0%) | 1 (16.7%) | 0 (0%) | 1 (20.0%) | 0 (0%) | 3 (9.4%) |
| | Epidural hematoma | 0 (0%) | 0 (0%) | 0 (0%) | 0 (0%) | 0 (0%) | 0 (0%) | 0 (0%) |
| | Subarachnoid hemorrhage | 0 (0%) | 0 (0%) | 1 (16.7%) | 0 (0%) | 0 (0%) | 0 (0%) | 1 (3.1%) |
| | Skull fracture | 6 (50.0%) | 0 (0%) | 1 (16.7%) | 0 (0%) | 2 (40.0%) | 0 (0%) | 9 (28.1%) |
| | Brain contusion | 0 (0%) | 0 (0%) | 0 (0%) | 0 (0%) | 0 (0%) | 0 (0%) | 0 (0%) |

(% of each factor in total of self-inflicted falls in each facility)

Self-inflicted falls were more common among boys in all facilities, and the average age ranged from 9 to 11 months. Parents/caregivers stated that 60–100% (mean 75.0%) of the injuries were caused by backward falls from a standing position, and 0–40% (mean 18.8%) were caused by backward falls from a sitting position. Retinal hemorrhages were present in 50–100% (mean 65.6%) and subdural hematomas in 58.3% to 100% (mean 78.1%). Skull fractures were observed in 0–50% of cases (mean 28.1%), and brain edemas were present in 0–16.7% (mean 9.4%, total 3 cases). There was only one case of subarachnoid hemorrhage and no cases of epidural hematoma or brain contusion.

The relationship between subdural hematoma, retinal hemorrhage and nonaccidental/accidental is shown in Table 6.

## Discussion

Retinal hemorrhage is often observed in abused children and is considered more likely to be abuse depending on its severity. This is thought to be caused by the retina being subjected to

**Table 6. The relationship between subdural hematoma, retinal hemorrhage and nonaccidental/accidental.**

| | SDH+RH | SDH only | RH only | no SDH,RH | Total |
|---|---|---|---|---|---|
| nonaccidental | 43 | 8 | 2 | 5 | 58 |
| accidental | 40 | 36 | 0 | 105 | 181 |

traction by the vitreous body when the head is shaken vigorously [1, 8]. Multivariate analysis in this study showed the odds ratio for retinal hemorrhage was 11.26 times higher in the non-accidental group.

Most of the nonaccidental cases in this study were associated with subdural hematoma.

There have been reports of shaken baby syndrome, in which retinal hemorrhage was observed despite the absence of intracranial findings [9–11], but in our study, only cases with some findings on CT or MRI were analyzed, so it is unclear to what extent retinal hemorrhage is observed in cases without intracranial findings.

The present data showed that self-inflicted falls significantly increased the odds of retinal hemorrhage to 6.22-fold. The data indicate that retinal hemorrhage occurs even with minor trauma, such as household trauma in infants. Several papers have reported that minor head injuries can result in retinal hemorrhage. In 1984, Aoki et al. reported a group of subdural and retinal hemorrhage cases due to mild head injury [12]. However, these cases had not been adequately investigated to determine whether abuse was involved [13]. Subsequently, there were several case reports of minor head injury with witnesses with a subdural hematoma and retinal hemorrhage. Aoki et al. reported a case of retinal hemorrhage due to a fall from a bed witnessed by an ICU nurse [14]. Atkinson et al. also reported eight cases of subdural hematoma due to backward falls, and all cases had retinal hemorrhage, although to varying degrees [15]. This report, and our findings may suggest that falls due to impact on the occipital region may be more morbid than other impact sites. What is curious about the results of the present study and previous reports is the high incidence of subdural hematoma and retinal hemorrhage due to self-inflicted falls rather than falls from other lower levels. In our study, the cases of self-inflicted falls with abnormalities on CT or MRI were characterized by boys around 9–11 months of age, with backward falls as the cause of injury, accompanied by subdural hematoma and retinal hemorrhage, but without brain edema or brain contusion. Self-inflicted falls were also found at all of the facilities in the study, suggesting that there is unlikely to be a bias in favor of a particular facility. We considered that many of the cases were not determined to be nonaccidental because many of the patients had no parenchymal brain damage and had a good prognosis. Furthermore, since we did not assess the severity of subdural hematomas or retinal hemorrhages in this study, it is possible that these factors were also milder in this group of patients. We do not deny that this result may be due to the lack of strict evaluation of abuse in Japan as a whole. The different percentages of patients identified as abusive in different facilities may be due to regional differences in abuse recognition. On the other hand, various reports from Japan pointed out that retinal hemorrhage and subdural hematoma occur in short fall, which is different from Western reports [16]. The presence of these cases may be unique to Japan. In fact, the cranial shape of infants differs between Westerners and Japanese [17]. In addition, head size varies by race [18] and in particular, Japanese breast-fed infants have larger head circumference than the world standard [19]. We assume that the size and shape of the cranium may change the impact of head trauma. Future research is needed to determine whether "self-inflicted falls" in this study were in fact caused by abuse or by accidental occipital impact.

Another question is, whether factors other than rotational-acceleration forces can cause retinal hemorrhage. Retinal hemorrhage can occur even when no strong external force is exerted directly on the eye. One factor is a sudden increase in intracranial pressure. There have been reports of retinal hemorrhage in infants with increased ICP, but it is reported that the retinal hemorrhage is mild, and the pattern is described as different from the retinal hemorrhage that occurs with abusive head trauma [20, 21]. Furthermore, several reports point out that in cases with severe retinal hemorrhage and intracranial hemorrhages such as subdural hematoma, retinal hemorrhage was not associated with the increased intracranial pressure but instead the

initial trauma [22, 23]. Birth-related retinal hemorrhage has been shown to occur frequently [24], especially in normal vaginal deliveries[25]. It is thought to occur when the infant head passes through a narrow parturient passage resulting in increased intracranial pressure, however this has not been proven [25].

In the present analysis, the intracranial findings associated with retinal hemorrhage were subdural hematoma and cerebral edema. In particular, a subdural hematoma was found in 97.6% of the patients with retinal hemorrhage and increased the odds by 23-fold, indicating a strong correlation. Studies have examined intracranial hemorrhage in animal models of retinal hemorrhage caused by an external force. Coats et al. examined the characteristics of ocular hemorrhages caused by a single rapid head rotation in neonatal pigs [26]. They also examined intracranial findings and found subdural hematomas in 43 of the 46 cases. These results indicate that the force that produce subdural hematoma are similar to those that produce retinal hemorrhage.

## Study limitations

In this study, cases temporarily taken into custody by the Child Guidance Center were classified as nonaccidental. Since whether the cases were later classified as accidental and their protection was lifted was not examined, it is possible that the number of nonaccidental cases was overestimated. Conversely, the high incidence of retinal hemorrhage due to minor head injury may indicate that nonaccidental cases is underestimated. In addition, only the presence or absence of retinal hemorrhage was considered for this study. Whether retinal hemorrhage was unilateral or bilateral, association with retinal schisis, or the severity of the hemorrhage was not considered. This needs to be addressed in future studies.

## Conclusion

Although nonaccidental, brain edema and self-inflicted falls were associated with retinal hemorrhage, the factor most strongly associated with retinal hemorrhage in the present study was subdural hematoma.

## Supporting information

**S1 Appendix. A list of the patients studied in this article with respect to age in months, gender, presence of retinal hemorrhage, presence of other intracranial lesions, the parent's stated injury history, and the final administrative determination of abuse.**
(DOCX)

## Author Contributions

**Conceptualization:** Masahiro Nonaka, Young-Soo Park.

**Data curation:** Mihoko Kato, Masahiro Nonaka, Nobuyuki Akutsu, Ayumi Narisawa, Atsuko Harada, Young-Soo Park.

**Formal analysis:** Mihoko Kato, Masahiro Nonaka.

**Investigation:** Masahiro Nonaka.

**Methodology:** Masahiro Nonaka, Young-Soo Park.

**Project administration:** Masahiro Nonaka.

**Resources:** Mihoko Kato, Masahiro Nonaka, Nobuyuki Akutsu, Ayumi Narisawa, Atsuko Harada, Young-Soo Park.

**Supervision:** Young-Soo Park.

**Writing – original draft:** Mihoko Kato, Masahiro Nonaka.

**Writing – review & editing:** Mihoko Kato, Masahiro Nonaka, Nobuyuki Akutsu, Ayumi Narisawa, Atsuko Harada, Young-Soo Park.

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
