## [Decision Letter · Decision Letter 0]

8 Jan 2023

PONE-D-22-34166Correlations of intracranial pathology and cause of head injury with retinal hemorrhage in infants and toddlers: A multicenter, retrospective study by the J-HITs (Japanese Head injury of Infants and Toddlers study) groupPLOS ONE

Dear Dr. Nonaka,

Thank you for submitting your manuscript to PLOS ONE. After careful consideration, we feel that it has merit but does not fully meet PLOS ONE’s publication criteria as it currently stands. Therefore, we invite you to submit a revised version of the manuscript that addresses the points raised during the review process.

We look forward to receiving your revised manuscript.

Kind regards,

Nader Hussien Lotfy Bayoumi, M.D., FRCS (Glasgow)

Academic Editor

PLOS ONE

Journal Requirements:

"Masahiro Nonaka and Young-Soo Park have written statements and appeared in court in child abuse cases both on the request of the prosecutor and the defense. Atsuko Harada has written statements and appeared in court in child abuse cases on the request of the prosecutor. This does not alter our adherence to PLOS ONE policies on sharing data and materials."

Reviewers' comments:

Reviewer's Responses to Questions

**Comments to the Author**

1. Is the manuscript technically sound, and do the data support the conclusions?

Reviewer #1: Partly

Reviewer #2: Yes

2. Has the statistical analysis been performed appropriately and rigorously? 

Reviewer #1: Yes

Reviewer #2: Yes

3. Have the authors made all data underlying the findings in their manuscript fully available?

Reviewer #1: No

Reviewer #2: Yes

4. Is the manuscript presented in an intelligible fashion and written in standard English?

Reviewer #1: No

Reviewer #2: Yes

5. Review Comments to the Author

Reviewer #1: The goals of this study are appropriate, but much needs clarification. Some of their reference literature is of very poor quality and some significant papers are missed.

I will just speak to the body of the manuscript, but that implies also a need to revise the abstract.

Methods, p3,L 15_ what is considered head trauma. This threshold for inclusion needs to be defined.

P4, L6-8 It's good to document both the caretaker's explanation and the cPT diagnosis.

P4, L13_ The authors need to provide information about the subjects excluded for lack of a retinal exam.

P4, L17_it's OK to include confessions as Dx of abuse, but to require that the kids were placed in temporary custody is the CPT suspected abuse is incorrect-the legal system is more flawed in it's abuse assessment. I'd just go with the CPT opinion.

P4, L24-"self-inflicted" needs more explanation and description.

The initial tables in results, beginning P5, L19-Although univariate ORs are later tabulated, it would help to see them in the initial tables.

Top table, P6-the authors need to state whether it is brain of scalp that was contused.

P, 6, L 3-See Dr Morad's 2 papers from 2004 in JAAPOS & recent Simon paper JAAPOS 2022 doi: 10/1016.j.jaapos.2022.10.003

P6, bottom table I note that with Major accidents: bicycle accidents, MVAs and falls > 2 M that far fewer kids have RH than children with Minor accidents: self-inflicted falls, bed or sofa falls and other falls < 2 M. This implies to me that the original data is incorrect. Most likely the children within the later groups were miss=diagnosed as not abuse. these are common false explanations. This might be an artifact of the authors failing to separate subjects into a few poster-pole, non-diagnostic RH vs many, to the periphery with or without schisis. This finding makes most of their conclusions suspect.

P9, L1-Though EDH is an impact injury and SAH can result from either impact or inertial injury. Kids with EDH can have inertial/rotational deceleration at impact. So it is unfair to conclude that impact injuries or "external forces" per se caused these injuries.

P 9, L3-4- seems entirely unsupported by their data and of unclear meaning-it should be deleted.

P9, L12-15 Gardner's case is also very suspect. It doesn't warrant inclusion as evidence. But Atkinson's series is reputable.

Top P 10. Terson's is very rare in infants. L6-7-the retinal vein also has acessory outflow paths that prevent it from causing retinal hypertension if more proximal obstruction Occurs_See Morad Clin & Exp Ophth 2010;38:514.

P 10, L11_ Shiau also discussed ICP Arch peds Adol Med 2012;166:623 & Minns Devel Med Child Neurol 2017;59:597 found that although intracranial hemorrhages such as SDH commonly occur in accidental and abusive TBI that the presence of severe RH was not associated with the ICP but instead the initial trauma.

P10, L19_ Thiblin's study can be discounted. he's a lousy and dishonest researcher-Greeley Acta Paediatrica 2022 doi: 10.1111/apa.16488, Stray-Pederson Acta Paediatrica 2022;11:793

P110. L 24- Saying that coats found iCP had a role in causing RH is incorrect, they only implicated indirect rotational acceleration forces. Theey didn't even look at ICP.

P11, L8: Although some abuse cases might be over-diagnosed in this study, I remain overwhelmingly concerned that much abuse still has been missed. se my comments re the Table on P6.

Above all I feel the authors are unjustified in their conclusions that RH result from the pressure from SDH and intra-cranial injuries as opposed to the same forces causing the SDH also causing the RH simultaneously and independently as a response to the forces experienced.

It's also problematic in this era to not separate minor and major RHs. Their specificity for abuse or major trauma is very different.

Reviewer #2: Accept, however figures are not provided

Good English language , clear message delivered . The statistics are satisfactory. The topic is interesting and applicable

6. PLOS authors have the option to publish the peer review history of their article (what does this mean?). If published, this will include your full peer review and any attached files.

Reviewer #1: **Yes: **Kenneth Feldman, MD

Reviewer #2: **Yes: **Karim A. Raafat MD.

---

## [Author Response · Author response to Decision Letter 0]

9 Feb 2023

Response to Reviewers

"Masahiro Nonaka and Young-Soo Park have written statements and appeared in court in child abuse cases both on the request of the prosecutor and the defense. Atsuko Harada has written statements and appeared in court in child abuse cases on the request of the prosecutor. This does not alter our adherence to PLOS ONE policies on sharing data and materials."

Author’s Response: There are no particular changes to be made. All data used in this study are disclosed as supplementary data in an anonymized form.

Author’s Response: The following sentences were added.

Change to Text: Page 5, Lines 6-10

IRB/ethics committee approval and a statement regarding patient consent

The protocol for this study was approved by the Ethics Committee of Kansai Medical University (No. 2019232). Need for written patient consent was waived by the ethics committee because data were deidentified. Institutional review board approval was obtained from all participants’ institutions prior to submitting cases for this study. 

1. Is the manuscript technically sound, and do the data support the conclusions?

Reviewer #1: Partly

Reviewer #2: Yes

2. Has the statistical analysis been performed appropriately and rigorously?

Reviewer #1: Yes

Reviewer #2: Yes

3. Have the authors made all data underlying the findings in their manuscript fully available?

Reviewer #1: No

Reviewer #2: Yes

Author’s Response: Please note that all data used in this study are disclosed as supplementary data in an anonymized form.

 4. Is the manuscript presented in an intelligible fashion and written in standard English?

Reviewer #1: No

Reviewer #2: Yes

5. Review Comments to the Author

Reviewer #1: The goals of this study are appropriate, but much needs clarification. Some of their reference literature is of very poor quality and some significant papers are missed.

1) I will just speak to the body of the manuscript, but that implies also a need to revise the abstract.

Author’s response: The abstract as well as the body of the manuscript has been changed.

2) Methods, p3,L 15_ what is considered head trauma. This threshold for inclusion needs to be defined. 

Author’s response: We defined head trauma in this paper as follows

Change to text: Page 3, Lines 13-16　

The definition of head trauma in our study included cases in which the parent or guardian stated during the interview that the child had suffered a head injury, as well as cases in which there were findings on imaging that were thought to be due to head trauma.　Furthermore, Patients with imaging findings, such as skull fractures or intracranial injuries, were included. Patients with no imaging findings, patients with only extracranial findings, and patients with no apparent findings on imaging were excluded.

3) P4, L6-8 It's good to document both the caretaker's explanation and the cPT diagnosis.

Author’s response: A new Table 3 summarizes the injury mechanism described by each guardian and which factors tend to be judged as abuse.

Change to text: Page 7, Lines 6-11 

Table 3 shows the injury mechanisms described by the parents in our study and the percentage of nonaccidental judgments for each. The presence or absence of retinal hemorrhages is also noted. The cases in which the parents did not fully explain the cause of the injury tended to be judged as abuse, while the cases in which the cause of the injury was fully explained tended to be judged as accidental. Retinal hemorrhage was present in 65.6% of self-inflicted falls.

Page 8, Lines 1-3 

Table 3.　Parents' description of the injury mechanism and determination of nonaccidental or accidental injury in this study.

4) P4, L13_ The authors need to provide information about the subjects excluded for lack of a retinal exam.

Author’s response:All 452 cases have already been reported. The present study is a detailed reanalysis of the presence/absence of retinal hemorrhage. References are listed below.

Change to text: Page 4, Line 12

 [1]

5) P4, L17_it's OK to include confessions as Dx of abuse, but to require that the kids were placed in temporary custody is the CPT suspected abuse is incorrect-the legal system is more flawed in it's abuse assessment. I'd just go with the CPT opinion.

Author’s response: In Japan, placement of temporary custody is implemented only by the decision of the Child Guidance Center, not by a court decision. Furthermore, In Japan, the CPT's opinion is not reported to the patient's physician for privacy purposes. So the CPT's decision is known to physicians only when the patient is taken into custody. 

6) P4, L24-"self-inflicted" needs more explanation and description.

Author’s response: We have explained about self-inflicted falls as follows

Change to text: Page 5, Lines 2-4

Self-inflicted falls occurs when an infant falls backward and bruises the back of the head, mainly when trying to stand while holding onto other objects. 

7) The initial tables in results, beginning P5, L19-Although univariate ORs are later tabulated, it would help to see them in the initial tables.

Author’s response: We have also included ORs in Tables 1 and 2 in response to the reviewer’s suggestion.

Change to text: Please see new Table 1 and 2

8) Top table, P6-the authors need to state whether it is brain of scalp that was contused.

Author’s response: The word “brain” was inserted before “contusion” to clarify this point.

Change to text: Please see new Table 1, 3, and 4

9) P, 6, L 3-See Dr Morad's 2 papers from 2004 in JAAPOS & recent Simon paper JAAPOS 2022 doi: 10/1016.j.jaapos.2022.10.003

Author’s response: We have cited the references given and added the following text

Change to text: Page 11, Lines 12-15 

There have been reports of shaken baby syndrome, in which retinal hemorrhage was observed despite the absence of intracranial findings {Morad, 2004 #170}{Morad, 2004 #171}{Simon, 2022 #175}, but in our study, only cases with some findings on CT or MRI were analyzed, so it is unclear to what extent retinal hemorrhage is observed in cases without intracranial findings.

10) P6, bottom table I note that with Major accidents: bicycle accidents, MVAs and falls > 2 M that far fewer kids have RH than children with Minor accidents: self-inflicted falls, bed or sofa falls and other falls < 2 M. This implies to me that the original data is incorrect. Most likely the children within the later groups were miss=diagnosed as not abuse. these are common false explanations. 

Author’s response: Among the minor injuries in the current study, self-inflicted falls were significantly associated with retinal hemorrhage. In order to examine whether our data are accurate, we have analyzed the frequency and characteristics of self-inflicted falls from each participating facility in Table 5. 

Change to text; Page 9, Lines 10-11, Page 10 , Lines 1-14 + Table 5

Self-inflicted falls, which were found to be significantly different in the multivariate analysis, have not been reported to be a factor associated with retinal hemorrhage. Therefore, we further examined the data in detail. Table 5 shows a detailed examination of the characteristics of self-inflicted falls by each facility. The results showed that self-inflicted falls were present in all facilities, ranging from 2.0-27.8% of the total number.

Self-inflicted falls were more common among boys in all facilities, and the average age ranged from 9 to 11 months. Parents/caregivers stated that 60-100% (mean 75.0%) of the injuries were caused by backward falls from a standing position, and 0-40% (mean 18.8%) were caused by backward falls from a sitting position. Retinal hemorrhages were present in 50-100% (mean 65.6%) and subdural hematomas in 58.3% to 100% (mean 78.1%). Skull fractures were observed in 0-50% of cases (mean 28.1%), and brain edemas were present in 0-16.7% (mean 9.4%, total 3 cases). There was only one case of subarachnoid hemorrhage and no cases of epidural hematoma or cerebral contusion.

Page 12, Lines 4-15

In our study, the cases of self-inflicted falls with abnormalities on CT or MRI were characterized by boys around 9-11 months of age, with backward falls as the cause of injury, accompanied by subdural hematoma and retinal hemorrhage, but without brain edema or brain contusion. Self-inflicted falls were also found at all of the facilities in the study, suggesting that there is unlikely to be a bias in favor of a particular facility. We considered that many of the cases were not determined to be nonaccidental because many of the patients had no parenchymal brain damage and had a good prognosis. We do not deny that this result may be due to the lack of strict evaluation of abuse in Japan as a whole. On the other hand, various reports from Japan pointed out that retinal hemorrhage and subdural hematoma occur in short fall, which is different from Western reports {Nonaka, 2022 #176}. The presence of these cases may be unique to Japan.

11) This might be an artifact of the authors failing to separate subjects into a few poster-pole, non-diagnostic RH vs many, to the periphery with or without schisis. This finding makes most of their conclusions suspect.

Author’s response: The intent of our study is to determine what factors influence retinal hemorrhage regardless of severity. So, we did not evaluate the severity of the retinal hemorrhage in this study. We added the following sentence in the limitation.

Change to text: Page 13, Lines 20-23 In addition, only the presence or absence of retinal hemorrhage was considered for this study. Whether retinal hemorrhage was unilateral or bilateral, or the severity of the hemorrhage was not considered. This needs to be addressed in future studies.

12) P9, L1-Though EDH is an impact injury and SAH can result from either impact or inertial injury. Kids with EDH can have inertial/rotational deceleration at impact. So it is unfair to conclude that impact injuries or "external forces" per se caused these injuries.

Author’s response: The following text has been deleted. “Still, some were associated with epidural hematoma or subarachnoid hemorrhage, so the possibility that an external force caused retinal hemorrhage to the head cannot be ruled out.”

Change to text: The deleted text was on Page 11, Line 10

13) P 9, L3-4- seems entirely unsupported by their data and of unclear meaning-it should be deleted.

Author’s response: The following sentence has been deleted. “However, it was unclear whether the force applied to the head needed to be as strong as an act of violence by the guardian to cause a retinal hemorrhage.”

Change to text: The deleted text was on Page 11, Line 10

14) P9, L12-15 Gardner's case is also very suspect. It doesn't warrant inclusion as evidence. But Atkinson's series is reputable.

Author’s response: We have removed the description of Gardner's paper. “Gardner reported a witnessed case of subdural hematoma with retinal hemorrhage due to a backward fall [9].”

Change to text: The deleted text was on Page 11, Line 23

15) Top P 10. Terson's is very rare in infants. L6-7-the retinal vein also has acessory outflow paths that prevent it from causing retinal hypertension if more proximal obstruction Occurs_See Morad Clin & Exp Ophth 2010;38:514. {Morad, 2010 #138}

Author’s response: We have removed the sentence about Terson. On the other hand, we described about birth-related retinal hemorrhage and the current possible mechanisms.

Change to text: Page 12 Line 24, Page 13, Lines 1-4 

Birth-related retinal hemorrhage has been shown to occur frequently {Watts, 2013 #177}, especially in normal vaginal deliveries{Cho, 2021 #178} . It is hypothesized to occur when the infant head passes through a narrow parturient passage resulting in increased intracranial pressure, however this has not been proven{Cho, 2021 #178}.

16) P 10, L11_ Shiau also discussed ICP Arch peds Adol Med 2012;166:623 & Minns Devel Med Child Neurol 2017;59:597 found that although intracranial hemorrhages such as SDH commonly occur in accidental and abusive TBI that the presence of severe RH was not associated with the ICP but instead the initial trauma.

Author’s response: We have cited the references given and added the following text

Change to text: Page 12, Lines 21-24

Furthermore, several reports point out that in cases with severe retinal hemorrhage and intracranial hemorrhages such as subdural hematoma, retinal hemorrhage was not associated with the increased intracranial pressure but instead the initial trauma[19,12].

17) P10, L19_ Thiblin's study can be discounted. he's a lousy and dishonest researcher-Greeley Acta Paediatrica 2022 doi: 10.1111/apa.16488, {Greeley, 2022 #173} Stray-Pederson Acta Paediatrica 2022;11:793 {Stray-Pedersen, 2022 #174}

Author’s response: In accordance with the Reviewer's comments, the following text has been removed. : 

“More recently, it has been reported that retinal hemorrhage may appear with a nonspecific cause associated with intracranial lesions such as subdural hematomas.” 

”Thiblin et al. compared the medical findings and types of trauma reported in infants with and without retinal hemorrhage. They found that intracranial pathology was significantly more common in retinal hemorrhage cases (97%) than in non-retinal hemorrhage cases (13%).” 

Change to text: The deleted text was on Page 3, Line 5, and Page 13, Line 8

18) P110. L 24- Saying that coats found iCP had a role in causing RH is incorrect, they only implicated indirect rotational acceleration forces. Theey didn't even look at ICP.

Author’s response: In response to the Reviewer's comments, we have revised the text as follows

Change to text: Page 13, Lines11-13 

Coats et al. examined the characteristics of ocular hemorrhages caused by a single rapid head rotation in neonatal pigs [7]. They also examined intracranial findings and found subdural hematomas in 43 of the 46 cases. These results indicate that the force that produce subdural hematoma are similar to those that produce retinal hemorrhage.

19) P11, L8: Although some abuse cases might be over-diagnosed in this study, I remain overwhelmingly concerned that much abuse still has been missed. se my comments re the Table on P6.

Author’s response: We specifically reanalyzed self-inflicted falls. We found that many of the cases had no brain parenchymal damage and probably had a good prognosis, despite the presence of retinal hemorrhage and subdural hematoma. This may be the reason why many of the cases were judged to have been accidents. Please see my response in 10). We also have added the following text to the Study limitation.

Change to text: Page 13, Lines 18-20

Conversely, the high incidence of retinal hemorrhage due to minor head injury may indicate that nonaccidental cases is underestimated.

20) Although some abuse cases might be over-diagnosed in this study, I remain overwhelmingly concerned that much abuse still has been missed.

Author’s response: Please see my response in 10).

21) Above all I feel the authors are unjustified in their conclusions that RH result from the pressure from SDH and intra-cranial injuries as opposed to the same forces causing the SDH also causing the RH simultaneously and independently as a response to the forces experienced.

Author’s response: We have removed the sentence “These results indicate that retinal hemorrhage is more likely to occur in association with subdural hematoma than with direct injury” from the abstract and the conclusion. 

Change to text: The deleted text was on Page 2. Line 15, and Page 14, Line 4

22) It's also problematic in this era to not separate minor and major RHs. Their specificity for abuse or major trauma is very different.

Author’s response: As the reviewer pointed out, the severity of retinal hemorrhage is very important to investigate the association with AHT.

However, we conducted this study because we believe that nothing can be done without first examining what factors cause retinal hemorrhage itself before examining the relationship with the severity of retinal hemorrhage.

Of course, based on our results, we think that in the future we need to conduct a multicenter study to examine the severity of retinal hemorrhage based on uniform criteria. We have added the following text to the Study limitation

Change to text: Page 13, Lines 20-23

In addition, only the presence or absence of retinal hemorrhage was considered, not whether it was unilateral or bilateral or the severity of the hemorrhage. This needs to be addressed in future studies.

Reviewer #2: Accept, however figures are not provided

Good English language , clear message delivered . The statistics are satisfactory. The topic is interesting and applicable

Author’s response: Thank you for the positive feedback and I will make sure to upload the Figures.

---

## [Decision Letter · Decision Letter 1]

27 Feb 2023

PONE-D-22-34166R1Correlations of intracranial pathology and cause of head injury with retinal hemorrhage in infants and toddlers: A multicenter, retrospective study by the J-HITs (Japanese Head injury of Infants and Toddlers study) groupPLOS ONE

Dear Dr. Nonaka,

Thank you for submitting your manuscript to PLOS ONE. After careful consideration, we feel that it has merit but does not fully meet PLOS ONE’s publication criteria as it currently stands. Therefore, we invite you to submit a revised version of the manuscript that addresses the points raised during the review process.

We look forward to receiving your revised manuscript.

Kind regards,

Nader Hussien Lotfy Bayoumi, M.D., FRCS (Glasgow)

Academic Editor

PLOS ONE

Journal Requirements:

Additional Editor Comments (if provided):

The paper is greatly improved, but still needs some minor work, partly to educate non-Japanese audiences and partly interpretive.

Page 3 or tracked changes document, line 21, I'd add:".....intracranial injuries of trauma who lacked caretaker histories of a traumatic event were included."

Page 4, L10-you indicate caretaker history was entered as the injury cause of the "final injury mechanism". Do you mean historical or your final judgment?

Page 4L13-115 and your response to reviewers point 5.: In that point you seem to suggest that if the Child Guidance Center is responsible for the decision whether to place the child in custody and only if this occurs is the referring providers notified of the abuse or not abuse judgment. I'm not sure I understand that correctly. But the text should document the actual practice. In you final diagnosis of abuse if you are your relying on caretaker history or the Child Guidance Center decision or something else. If the CGC, doesn't this imply you are naive to some decisions of whether abuse occurred or not. This whole section needs to be revised to educate the reader's about the Japanese process and what you are using when you say a child was abused or not.

In Table 2, P7: I remain concerned that self-inflicted falls, falling from a bed or sofa and other < 2 m falls seem to have more morbid consequences than falls for greater heights, MVAs and bicycle accidents. The discussion on P12, l22-23 may partly explain this in that it is indicated that those kids with SDH and RH often lacked brain edema or contusion.

Kids with simple falls often have small subjacent contact SDHs, but lack accompanying brain injury and could have a few minor/non-specific RH. Unfortunately, the data to decide whether this is the explanation is lacking as RH were binomally present or absent and more concerning SDHs were not separated from simple subjacent contact injuries. This confolict with reality will need more elaboration in the discussion.

Also, adding Table 5 helps a little to reassure readers that those minor injury mechanisms were seen at all participating institutions, as the authors noted. But the authors did not comment that hospital A saw 6.7% were abusive and hospital C 66.7% abusive, but the other 4 found no abuse cases among these minor trauma events. This reinforces my original opinion that although Japanese medicine is doing better at recognizing that abuse occurs, that remains very spotty. And at some institutions, fabricated history is usually accepted at face vale, while some are doing better. Those %s of abuse diagnosis need to be addressed in the discussion (in the vicinity of page 12, line 20-24) and comments are needed about the real possibility that abuse is still frequently remaining undiagnosed.

Page 10, L10: It is interesting that the minor injury event kids had had a lot of skull Fxs, but no EDHs or brain contusions. Both are commonly present with primarily impact events that cause skull Fxs at the same time.

Somewhere in Results I'd like to have reported. how many abused vs non-abused kids had SDH + RH vs SDH - RH and RH _ SDH vs RH - SDH.

P 13, L5; I think it is important to recognize that Japanese children are highly unlikely to be uniquely susceptible to SDHs from minor falls. Obviously this would be racially stereotyping, when race is a poor proxy for anything biologic. If it were true, we would expect Japanese children to experience the same risk if they lived elsewhere, say the US where they sustain fewer injuries than others. We also would expect the same brain injury susceptibility to extend to other cause of TBI, say falls from significant heights or MVAs, but the current data don't support that. I expect this is a hold over form the era when it was culturally unacceptable to admit that abuse occurred in Japan (all Aoki's early writing). I think these are ideas that have long over-lived their time and should be buried by more critical case reviews.

It would be worth adding that these is at least a case series to suggest falls with occipital impacts may be more morbid than with other impact sites. Atkinson Pediatric Emerg care 2018;34:837.

P 15, L7: I appreciate the addition of the discussion the NAT may be underestimated. The following statements could be edited to more clearly state that no dividing RH into type number, location, severity in some way precludes conclusions about minor vs more specific complex, profuse RH patterns with or without schisis.

I think it would be OK in the conclusion to add that brain edema in multi-variable analysis was also associated with RH (see Binenbaum Pediatrics. 2013;132:e430) . And a sentence saying that the association of RH with SDH & brain edema can't imply the SDH caused the RH vs the same forces that caused the SDH, brain edema.

Reviewers' comments:

Reviewer's Responses to Questions

**Comments to the Author**

1. If the authors have adequately addressed your comments raised in a previous round of review and you feel that this manuscript is now acceptable for publication, you may indicate that here to bypass the “Comments to the Author” section, enter your conflict of interest statement in the “Confidential to Editor” section, and submit your "Accept" recommendation.

Reviewer #1: (No Response)

Reviewer #2: All comments have been addressed

2. Is the manuscript technically sound, and do the data support the conclusions?

Reviewer #1: Partly

Reviewer #2: Yes

3. Has the statistical analysis been performed appropriately and rigorously? 

Reviewer #1: Yes

Reviewer #2: Yes

4. Have the authors made all data underlying the findings in their manuscript fully available?

Reviewer #1: Yes

Reviewer #2: Yes

5. Is the manuscript presented in an intelligible fashion and written in standard English?

Reviewer #1: (No Response)

Reviewer #2: Yes

6. Review Comments to the Author

Reviewer #1: The paper is greatly improved, but still needs some minor work, partly to educate non-Japanese audiences and partly interpretive.

Page 3 or tracked changes document, line 21, I'd add:".....intracranial injuries of trauma who lacked caretaker histories of a traumatic event were included."

Page 4, L10-you indicate caretaker history was entered as the injury cause of the "final injury mechanism". Do you mean historical or your final judgment?

Page 4L13-115 and your response to reviewers point 5.: In that point you seem to suggest that if the Child Guidance Center is responsible for the decision whether to place the child in custody and only if this occurs is the referring providers notified of the abuse or not abuse judgment. I'm not sure I understand that correctly. But the text should document the actual practice. In you final diagnosis of abuse if fyou are your relying on caretaker history or the Child Guidance Center decision or something else. If the CGC, doesn't this imply you are naive to some decisions of whether abuse occurred or not. This whole section needs to be revised to educate the reader's about the Japanese process and what you are using when you say a child was abused or not.

In Table 2, P7: I remain concerned that self-inflicted falls, falling from a bed or sofa and other < 2 m falls seem to have more morbid consequences than falls fro greater heights, MVAs and bicycle accidents. The discussion on P12, l22-23 may partlly explain this in that it is indicated that those kids with SDH and RH often lacked brain edema or contusion.

Kids with simple falls often have small subjacent contact SDHs, but lack accompanying brain injury and could have a few minor/non-specific RH. Unfortumately, the data to decide whether this is the explanation is lacking as RH were binomally present or absent and more concerning SDHs were not separated form simple subjacent contact injuries. This confo\\ict with reality will need more elaboration in the discussion.

Also, adding Table 5 helps a little to reassure readers that those minor injury mechanisms were seen at all participting institutions, as the authors noted. But the did not comment that hospital A saw 6.7% were abusive and hospital C 66.7% abusive, but the other 4 found no abuse cases among these minor trauma events. This reinforces my original opinion that although Japanese medicine is doing better at recognizing that abuse occurs, that remains very spotty. And at some institutions, fabricated history is usually accepted at face vale, while some are doing better. Those %s of abuse diagnosis need to be addressed in the discussion ( in the vicinity of page 12, line 20-24) and comments are needed about the real possibility that abuse ifs still frequently remaining undiagnosed.

Page 10, L10: It is interesting that the minor injury event kids had had a lot of skull Fxs, but no EDHs or brain contusions. Both are commonly present with primarily impact events that cause skull Fxs at the same time.

Somewhere in Results I'd like to have reported. how many abused vs non-abused kids had SDH + RH vs SDH - RH and RH _ SDH vs RH - SDH.

P 13, L5; I think it is important to recognize that Japanese children are highly unlikely to be uniquely susceptible to SDHs from minor falls. Obviously this would be racially stereotyping, when race is a poor proxy for anything biologic. If it were true, we would expect Japanese children to experience the same risk if they lived elsewhere, say the US where they sustain fewer injuries than others. We also would expect the same brain injury susceptabilty to extend to other cause of TBI, say falls from significant heights or MVAs, but the current data don't support that. I expect this is a hold over form the era when it was culturally unacceptable to admit that abuse occurred in Japan (all Aoki's early writing). I think these are ideas that have long over-lived their time and should be buried by more critical case reviews.

It would be worth adding that these is at least a case series to suggest falls with occipital impacts may be more morbid than with other other impact sites. Atkinson Pediatric Emerg care 2018;34:837.

P 15, L7: I appreciate the addition of the discussion the NAT may be underestimated. The following statements could be edited to more clearly state that no dividing RH into type number, location, severity in some way precludes conclusions anbout minor vs more specific complex, profuse RH patterns with or without schisis.

I think it would be OK in the conclusion to add that brain edema in multi-variable analysis was also associated with RH (see Binenbaum Pediatrics. 2013;132:e430) . And a sentence saying that the association of RH with SDH & brain edema can't imply the SDH caused the RH vs the same forces that caused the SDH, brain edema.

Reviewer #2: (No Response)

7. PLOS authors have the option to publish the peer review history of their article (what does this mean?). If published, this will include your full peer review and any attached files.

Reviewer #1: **Yes: **Kenneth W. Feldman, MD

Reviewer #2: **Yes: **Karim Raafat

---

## [Author Response · Author response to Decision Letter 1]

2 Mar 2023

Response to Reviewers

Reviewer #1: The paper is greatly improved, but still needs some minor work, partly to educate non-Japanese audiences and partly interpretive.

Author’s Response: We thank the reviewer for valuable comments.

１）Page 3 or tracked changes document, line 21, I'd add:".....intracranial injuries of trauma who lacked caretaker histories of a traumatic event were included."

Author’s Response: The following sentences were added.

Change to text: Page 3, Lines 17-18

intracranial injuries of trauma who lacked caretaker histories of a traumatic event were included

2) Page 4, L10-you indicate caretaker history was entered as the injury cause of the "final injury mechanism". Do you mean historical or your final judgment?

Author’s Response: Since the cause of the injury described by the guardian may change, it means that the most recent description of how the injury occurred was recorded. To make this point clear, the sentence was rewritten.

Change to text: Page 4, Lines 9-10

If there were any changes in the parents’ explanation of the cause of the injury, they were noted verbatim, and the most recent description of the injury, as stated by the parents, was included in the study.

3) Page 4L13-115 and your response to reviewers point 5.: In that point you seem to suggest that if the Child Guidance Center is responsible for the decision whether to place the child in custody and only if this occurs is the referring providers notified of the abuse or not abuse judgment. I'm not sure I understand that correctly. But the text should document the actual practice. In you final diagnosis of abuse if fyou are your relying on caretaker history or the Child Guidance Center decision or something else. If the CGC, doesn't this imply you are naive to some decisions of whether abuse occurred or not. This whole section needs to be revised to educate the reader's about the Japanese process and what you are using when you say a child was abused or not.

Author’s Response: The following sentences were added.

Change to text: Page 5, Lines 1-12

The Childs Guidance Center in Japan receives notifications of suspected cases of abuse at medical institutions, etc., and determines whether or not abuse actually occurred. The CGC determines whether or not the child should be placed in temporary custody based on the child's history of visits to medical institutions and health checkups, whether or not the sibling has been abused, and the results of a diagnosis by a third-party doctor. The process and reasons for how the determination of whether or not abuse was made will not be communicated to the medical institution that notified of the abuse. In cases where abuse is strongly suspected, the child is almost always given temporary protection. On the other hand, cases that are determined to be unlikely to be abusive are not protected. Therefore, in this study, the cases that were temporarily protected by CGC were judged as abuse, and those that were not protected were judged as accidents.

5) In Table 2, P7: I remain concerned that self-inflicted falls, falling from a bed or sofa and other < 2 m falls seem to have more morbid consequences than falls fro greater heights, MVAs and bicycle accidents. The discussion on P12, l22-23 may partlly explain this in that it is indicated that those kids with SDH and RH often lacked brain edema or contusion.

Kids with simple falls often have small subjacent contact SDHs, but lack accompanying brain injury and could have a few minor/non-specific RH. Unfortumately, the data to decide whether this is the explanation is lacking as RH were binomally present or absent and more concerning SDHs were not separated form simple subjacent contact injuries. This confo\\ict with reality will need more elaboration in the discussion.

Author’s Response: The following sentences were added.

Change to text: Page 13, Lines 14-16

Furthermore, since we did not assess the severity of subdural hematomas or retinal hemorrhages in this study, it is possible that these factors were also milder in this group of patients.

6) Also, adding Table 5 helps a little to reassure readers that those minor injury mechanisms were seen at all participting institutions, as the authors noted. But the did not comment that hospital A saw 6.7% were abusive and hospital C 66.7% abusive, but the other 4 found no abuse cases among these minor trauma events. This reinforces my original opinion that although Japanese medicine is doing better at recognizing that abuse occurs, that remains very spotty. And at some institutions, fabricated history is usually accepted at face vale, while some are doing better. Those %s of abuse diagnosis need to be addressed in the discussion ( in the vicinity of page 12, line 20-24) and comments are needed about the real possibility that abuse ifs still frequently remaining undiagnosed.

Author’s Response: The following sentences were added.

Change to text: Page 13, Lines 17-19

The different percentages of patients identified as abusive in different facilities may be due to regional differences in abuse recognition.

7) Page 10, L10: It is interesting that the minor injury event kids had had a lot of skull Fxs, but no EDHs or brain contusions. Both are commonly present with primarily impact events that cause skull Fxs at the same time.

Author’s Response: It is true that there were many cases with only skull fractures and no intracranial abnormalities. However, since this is not the focus of the paper, we will not discuss about this in the text.

8) Somewhere in Results I'd like to have reported. how many abused vs non-abused kids had SDH + RH vs SDH - RH and RH _ SDH vs RH - SDH.

Author’s Response: We have created a new Table.

Change to text: Page 11, Lines 10-13 + Table 6 is added.

The relationship between subdural hematoma, retinal hemorrhage and nonaccidental/accidental is shown in Table ６.

9) P 13, L5; I think it is important to recognize that Japanese children are highly unlikely to be uniquely susceptible to SDHs from minor falls. Obviously this would be racially stereotyping, when race is a poor proxy for anything biologic.

If it were true, we would expect Japanese children to experience the same risk if they lived elsewhere, say the US where they sustain fewer injuries than others.

We also would expect the same brain injury susceptabilty to extend to other cause of TBI, say falls from significant heights or MVAs, but the current data don't support that.

Author’s Response: We consider that differences by race and culture cannot be ignored. It is also true that the cranial shape of infants differs between Westerners and Japanese, and that head size differs by race, especially for Japanese breast-fed infants, who are larger than the global standard. As you know, macrocephaly is prone to subdural hematoma.

We are also concerned that Japanese or Asian cases of subdural hematoma combined with retinal hemorrhage in countries other than Japan, especially in the U.S, may not be reported in the scientific journal because medical institutions and /or public agencies may automatically diagnose the cause of the injury as abuse. Gardner's paper which we removed from citation according to the reviewer's suggestion, was about Asian infant.

Change to text: Page 13, Lines 21-24, Page 14, Line 1

In fact, the cranial shape of infants differs between Westerners and Japanese {Koizumi, 2010 #179}. In addition, head size varies by race {Natale, 2014 #182} and in particular, Japanese breast-fed infants have larger head circumference than the world standard. {Tanaka, 2013 #181}　 We assume that the size and shape of the cranium may change the impact of head trauma.

10) I expect this is a hold over form the era when it was culturally unacceptable to admit that abuse occurred in Japan (all Aoki's early writing). I think these are ideas that have long over-lived their time and should be buried by more critical case reviews.

Author’s Response: We believe that we need to fully examine one idea before burying it. The following sentences were added.

Change to text: Page 14, Lines 2-3

Future research is needed to determine whether “self-inflicted falls” in this study were in fact caused by abuse or by accidental occipital impact.

11) It would be worth adding that these is at least a case series to suggest falls with occipital impacts may be more morbid than with other other impact sites. Atkinson Pediatric Emerg care 2018;34:837.

Author’s Response: The following sentences were added.

Change to text: Page 13, Lines 3-5

This report, and our findings may suggest that falls due to impact on the occipital region may be more morbid than other impact sites.

P 15, L7: I appreciate the addition of the discussion the NAT may be underestimated. The following statements could be edited to more clearly state that no dividing RH into type number, location, severity in some way precludes conclusions anbout minor vs more specific complex, profuse RH patterns with or without schisis.

Author’s Response: The following sentences were added.

Change to text: Page 15, Lines 11-13

Whether retinal hemorrhage was unilateral or bilateral, association with retinal schisis, or the severity of the hemorrhage was not considered.

I think it would be OK in the conclusion to add that brain edema in multi-variable analysis was also associated with RH (see Binenbaum Pediatrics. 2013;132:e430) . And a sentence saying that the association of RH with SDH & brain edema can't imply the SDH caused the RH vs the same forces that caused the SDH, brain edema.

Author’s Response: Thank you for your advice. However, we did not make any changes because we thought the current conclusion was simpler and better.

---

## [Decision Letter · Decision Letter 2]

6 Mar 2023

Correlations of intracranial pathology and cause of head injury with retinal hemorrhage in infants and toddlers: A multicenter, retrospective study by the J-HITs (Japanese Head injury of Infants and Toddlers study) group

PONE-D-22-34166R2

Dear Dr. Nonaka,

We’re pleased to inform you that your manuscript has been judged scientifically suitable for publication and will be formally accepted for publication once it meets all outstanding technical requirements.

Kind regards,

Nader Hussien Lotfy Bayoumi, M.D., FRCS (Glasgow)

Academic Editor

PLOS ONE

Additional Editor Comments (optional):

The Editor thanks the authors for their swift and concise responses to the reviewers' comments.

Reviewers' comments:

Reviewer's Responses to Questions

**Comments to the Author**

1. If the authors have adequately addressed your comments raised in a previous round of review and you feel that this manuscript is now acceptable for publication, you may indicate that here to bypass the “Comments to the Author” section, enter your conflict of interest statement in the “Confidential to Editor” section, and submit your "Accept" recommendation.

Reviewer #1: All comments have been addressed

2. Is the manuscript technically sound, and do the data support the conclusions?

Reviewer #1: Yes

3. Has the statistical analysis been performed appropriately and rigorously? 

Reviewer #1: Yes

4. Have the authors made all data underlying the findings in their manuscript fully available?

Reviewer #1: Yes

5. Is the manuscript presented in an intelligible fashion and written in standard English?

Reviewer #1: Yes

6. Review Comments to the Author

Reviewer #1: I appreciate the authors' efforts to respond to the queries. Many of the points that were unclear due to differences in protective practices are now explained. I remain concerned by the explanations about differences between Asian and other "Racial" groups. Even if head sizes are on the whole larger, it's not larger head sizes that predispose to ICH, but expanded subarachnoid spaces. I agree internal skull configurations could possibly change some susceptibilities. However, in the future I suspect that as Japanese skills and diligence diagnosing abuse improves, we will find that their infants are pretty much like any other. I don't think this worth quibbling about this any more or holding the paper up for another round of edits. I'm willing to accept the current efforts.

7. PLOS authors have the option to publish the peer review history of their article (what does this mean?). If published, this will include your full peer review and any attached files.

Reviewer #1: **Yes: **Kenneth Feldman, MD

---

## [Editor Report · Acceptance letter]

10 Mar 2023

PONE-D-22-34166R2 

Correlations of intracranial pathology and cause of head injury with retinal hemorrhage in infants and toddlers: A multicenter, retrospective study by the J-HITs (Japanese Head injury of Infants and Toddlers study) group 

Dear Dr. Nonaka:

I'm pleased to inform you that your manuscript has been deemed suitable for publication in PLOS ONE. Congratulations! Your manuscript is now with our production department. 

Kind regards, 

on behalf of

Professor Nader Hussien Lotfy Bayoumi 

Academic Editor

PLOS ONE